# Fatty-Acid-Rich *Agave angustifolia* Fraction Shows Antiarthritic and Immunomodulatory Effect

**DOI:** 10.3390/molecules27217204

**Published:** 2022-10-24

**Authors:** Enrique Jiménez-Ferrer, Gabriela Vargas-Villa, Gabriela Belen Martínez-Hernández, Manases González-Cortazar, Alejandro Zamilpa, Maribel Patricia García-Aguilar, Martha Lucía Arenas-Ocampo, Maribel Herrera-Ruiz

**Affiliations:** 1Centro de Investigación Biomédica del Sur, Instituto Mexicano del Seguro Social, Argentina No. 1, Col. Centro, Xochitepec 62790, Morelos, Mexico; 2Centro de Desarrollo de Productos Bióticos, Instituto Politécnico Nacional, Col. San Isidro, Yautepec 62739, Morelos, Mexico

**Keywords:** *Agave angustifolia*, edible plant, fatty acids, rheumatoid arthritis, pain, inflammation, cytokines

## Abstract

*Agave angustifolia* is a xerophytic species widely used in Mexico as an ingredient in sweet food and fermented beverages; it is also used in traditional medicine to treat wound pain and rheumatic damage, and as a remedy for psoriasis. Among the various *A. angustifolia* extracts and extract fractions that have been evaluated for their anti-inflammatory effects, the acetonic extract (AaAc) and its acetonic (F-Ac) and methanolic (F-MeOH) fractions were the most active in a xylene-induced ear edema model in mice, when orally administered. Four fractions resulting from chemically resolving F-Ac (F1–F4) were locally applied to mice with phorbol 12-myristate 13-acetate (TPA)-induced ear inflammation; F1 inhibited inflammation by 70% and was further evaluated in a carrageenan-induced mono-arthritis model. When administered at doses of 12.5, 25, and 50 mg/kg, F1 reduced articular edema and the spleen index. In addition, it modulated spleen and joint cytokine levels and decreased pain. According to a GC–MS analysis, the main components of F1 are fatty-acid derivatives: palmitic acid methyl ester, palmitic acid ethyl ester, octadecenoic acid methyl ester, linoleic acid ethyl ester, and oleic acid ethyl ester.

## 1. Introduction

*Agave,* a Latin term that means “something great, illustrious, dignified”, is the name Carl Linnaeus assigned to various species [1]. *Agave*, a specific genus in the Agavaceae family, has been used historically as a source of fiber, as an ingredient in beverages and food, to produce clothing, as fodder, as an ornament, as a building material, and as a remedy. The genus has approximately two hundred species, 75% of which are distributed in Mexico, and this country is regarded as the center of origin of these plants, since about 50% of the species are endemic [2,3]. Agaves were of great economic importance to Mesoamerican cultures for nine thousand years [4], and some of their pre-Hispanic uses are still alive today, e.g., as a source of fiber; for the consumption of their sap (aguamiel); for alcoholic fermentation; and the use of the plant’s core (piña) and stems as food, fodder, and for medicinal purposes [1]. *Agave angustifolia* Haw var. *angustifolia*, an *Agave* subgenus in the Rigidae group, commonly called “maguey espadín” or “maguey de monte” [5], is widely distributed in Mexico, from Sonora and Chihuahua to the south of the country, and as far as Nicaragua. It grows in a wide range of ecosystems: coastal dunes, low deciduous forest, the margins of medium sub-deciduous forests, some types of xerophytic scrub, palm groves, and oak–pine forests. This species lives both at sea level and above an altitude of 2000 m. It has fibrous, thin, narrow leaves, with small teeth on the margins and a dark-colored terminal spine; its color varies from gray to olive green; its inflorescence measures 2–4 m, with open branches, yellow-green flowers, and abundant nectar; and once pollinated, it produces a fruit with black seeds [3,5]. Its fibrous stems are used to make rope, twine, and bags; both its flowers and stems are edible and are often used to make traditional foods and sweets, and its sap is consumed either raw or fermented [3]. The uses of *A. angustifolia* in traditional medicine are diverse. The fiber is used to treat urticaria; the sap, cooked leaves, and root infusions have been used for “internal injuries”; a concoction of the root has been used against dysentery; fresh leaves are useful to stop bleeding and alleviate wound pain, to treat skin pimples, and for coughs; and roasted leaves are used to relieve rheumatic pain and as a remedy for sprains or broken bones. Interestingly, the plant has also been used to treat psoriasis, an autoimmune disease that causes joint pain and inflammation [6,7]. Several studies on the pharmacologic activity of *A. angustifolia* have been conducted. Ethanolic extract from the leaves of this agave allowed the identification of its antioxidant capacity in vitro using DPPH, ABTS, and FRAP methods; the same treatment also demonstrated antimicrobial activity against *Staphylococcus epidermidis* and *Escherichia coli* [8]. The administration o.p. of a hydroethanolic extract of the leaf of *A. angustifolia* to Wistar rats decreased the presence of gastric ulcers induced with absolute ethanol by 90% [9]. It has been shown that *A. angustifolia* has effects on metabolism; for example, the administration of fructans of this plant to a group of healthy rats and another group of diabetic animals showed a decrease in the blood concentration of cholesterol and LDL proteins, and hepatic steatosis was observed in the latter group; in addition, the fecal *Lactobacillus* spp. and *Bifidobacterium* spp. counts were greater than in those who did not receive fructans [10]. Furthermore, the incorporation of agavinas, which are also considered a source of dietary fiber, into the diet of mice caused beneficial effects by increasing the concentration of glucagon-like peptide-1 (GLP-1) and decreasing ghrelin (involved in the regulation of eating behavior) [11].

The medicinal use of and biological studies on *A. angustifolia* led to research into its effect on rheumatoid arthritis (RA), which is an autoimmune inflammatory disease characterized by aggressive synovial hyperplasia, resulting in joint destruction manifested by pain, swelling, and stiffness, along with bone erosion and rheumatoid nodules under the skin. Fatigue; weight loss; and extra-articular disorders involving the eyes, oral cavity, blood, lungs, skin, heart, kidneys, nerves, and lymph nodes may also be observed [12]. The World Health Organization (WHO) has reported a prevalence of 0.3–1% for RA; this condition, which is more common in women and in developed countries, contributes 14 million patients to the overall burden of musculoskeletal conditions worldwide [13]. In Mexico, a prevalence of 1.6% was determined for RA in the general population. The disease has a significant impact on the most productive age group, resulting in high rates of work incapacity that affect the economy and quality of life of patients [14,15]. In 2018, in México, RA was a significant health problem in the population aged 50 years and over, with a female/male ratio of 3:1 [16].

The joints of patients with RA suffer from the constant and sustained entry of immune cells, which form a vicious circuit in the promotion and release of proinflammatory mediators, such as cytokines, which trigger the activation of fibroblast-like synoviocytes, contributing to the bone and cartilage damage. Then, the deregulation in the release of cytokines plays an essential role in the pathology of RA; one of the primary sources of these molecules is macrophages. In mice, two types of these synovial cells are recognized, intrinsic macrophages present in the joint synovium from birth and extrinsic bone-marrow-derived macrophages. The first express the anti-inflammatory cytokines IL-10 and IL-4, the latter the proinflammatory cytokines, such as tumor necrosis factor-alpha (TNF-α) and interleukin (IL)-1β [17]. TNF is a crucial cytokine acting as a regulator in the pathogenesis of this disease, and its expression is increased in patients with RA. It activates endothelial cells and recruits synovial fibroblasts and macrophages to release IL-1β, IL-6, and TNF [18,19]. IL-17 is a pro-inflammatory cytokine that controls osteoclast differentiation and antibody increase, together with TNF-a and IL-6. IL-17 participates in the initial stages, and when the disease is already established, its overexpression correlates with clinical parameters of RA; this cytokine promotes the activation of fibroblast-like synoviocytes and the recruitment of neutrophils, macrophages, and B cells [20,21]. Another inflammatory marker of the cytokine family is IL-1β, produced mainly by monocytes and macrophages, but also by other types of cells, such as synovial cells. In turn, this molecule activates monocytes/macrophages and increases the proliferation of fibroblasts, prolonging and increasing the inflammation of the synovial membrane. In addition, it induces the activation of chondrocytes and osteoclasts, which causes cartilage damage and bone resorption [22]. In this context of joint damage potentiated by the action of proinflammatory cytokines, IL-10 is described as a potent immunomodulator that inhibits neutrophil infiltration and synovial tissue activation. Immune modulation is fundamental, and cytokine IL-10 actively participates in this line of action, due to its potent ability to regulate the immune response by inhibiting the infiltration of neutrophils and the activation of synovial tissue. In addition, this molecule promotes the polarization of macrophages towards an M2 phenotype and inhibits the expression of TNF [23,24].

RA treatment usually starts with non-steroidal anti-inflammatory drugs (NSAIDs) such as paracetamol, celecoxib, indomethacin, naproxen, and diclofenac to alleviate pain and inflammation. Since NSAIDs do not address the cause of the disease, they are accompanied by disease-modifying drugs (DMARDs) such as methotrexate (MTX), sulfasalazine, and azathioprine; in a more recent approach, biologics such as anti-TNF, abatacept, and rituximab have been used [25].

In searching for valuable therapies to improve patients’ quality of life with fewer adverse effects, the use of medicinal plants is an attractive approach. Considering the wide use of *A. angustifolia* in Mexico and its medicinal properties, including its reported efficacy in inflammatory and autoimmune disorders, this work aimed to evaluate the anti-inflammatory effects of extracts and fractions from this plant in murine models of xylene and ear edema, as well as its immunomodulatory activity in carrageenan/kaolin (CK)-induced mono-arthritis through the quantification of the relevant cytokines, such as TNF-α, IL-1α, IL-17, and IL-10.

## 2. Results

### 2.1. Chemical Analysis

One kilogram of lyophilized product was obtained from fresh *A. angustifolia* Haw leaves (4.3 kg). After a first extraction of the leaves with acetone, 35 g of product (AaAc) was obtained, providing a yield of 3.5%. Then, 4.9 g of AaAc was fractioned, with yields of 6.4% (0.31 g) for the n-hexane fraction (F-Hex), 68.12% (3.33 g) for the ethyl acetate fraction (F-AcOEt), 20.87% (1.02 g) for the acetonic fraction (F-Ac), and 4.61% (0.22 g) for the methanolic fraction (F-MeOH). F-Ac was sub-fractioned, with yields of 8.6% (88 mg) for F1, 5.6% (58 mg) for F2, 3.13% (32 mg) for F3, and 2.54% (25 mg) for F4.

### 2.2. Anti-Inflammatory Activity of A. angustifolia Extracts

Local xylene application to the ears of mice induced edema, resulting in a weight difference of 8.87 mg in the vehicle-administered group (VEH). Edema weight was reduced by 72% after the oral administration of indomethacin (INDO). Treating mice with AaAc at a dose of 200 mg/kg also inhibited ear edema with respect to the VEH group (* *p* < 0.05). While all extract fractions caused a significant reduction in edema weight with respect to the VEH group, F-MeOH and F-Ac showed the highest inhibitory effect (Table 1).

To evaluate the effect of the F-MeOH-derived fractions, the TPA (phorbol 12-myristate 13-acetate)-induced edema assay was used. As shown in Table 2, TPA-induced edema resulted in a weight difference of 10.24 mg in vehicle-treated mice. This effect was inhibited by INDO and F1 at a dose of 1.0 mg/ear. While F2, F3, and F4 also induced a statistically significant effect, the percentage of inhibition was less than 40%.

### 2.3. Effect of A. angustifolia Extract on Mono-Arthritis Induced by K/C

Based on the results of the inhibition of TPA-induced edema, the anti-inflammatory activity of F1 was assessed at three different doses in a mono-arthritis model. The area under the curve for joint inflammation (mm) 24 h after the administration of either K/C or SSF (basal) is shown in Figure 1. The positive control drug, methotrexate, reduced inflammation levels for the duration of the experiment. At the same time, F1 failed to modify K/C-induced inflammation at the lowest dose, but it reduced articular edema at doses of 50 mg/kg (* *p* < 0.05).

As shown in Figure 2, the time elapsed before mouse response on the hot plate test was significantly shorter in K/C-treated than in SS-treated animals (* *p* < 0.05). This effect was offset by the administration of F1 at doses of 25 and 50 mg/kg (* *p* < 0.05).

Spleen size, expressed as a percentage of total body weight, increased in animals treated with K/C with respect to SS-treated mice. Both methotrexate and F1 at doses of 12.65 and 50 mg/kg significantly reduced this effect with respect to the VEH group (* *p* < 0.05, Table 3). Interestingly, an intermediate dose of F1 failed to revert spleen growth.

#### 2.3.1. Effect of *A. angustifolia* Extract on Cytokine Concentration in Organs from K/C-Treated Mice

Cytokine levels were measured in the spleen and left knee joint of mice. As shown in Table 4, K/C-treated animals showed increased levels of IL-1, TNF-α, and IL-17 in the spleen, whereas IL-10 levels were significantly decreased with respect to SS-treated control animals (* *p* < 0.05). A significant decrease in TNF-α levels and an increase in IL-10 levels were observed in all treatment groups; however, while a decrease in IL-1 levels was observed in all groups with respect to the VEH group, it was significant only for mice treated with F1 at doses of 12.5 and 25 mg/kg (* *p* < 0.05). Finally, no significant changes in spleen IL-17 levels were observed under any treatment.

On the other hand, TNF-α and IL-17 levels in the joint were increased in mice in the VEH group, while the IL-10 concentration was much lower with respect to SS-treated animals (* *p* < 0.05). Both methotrexate and F1 significantly reduced these effects, and the reduction was greater when F1 was administered at a dose of 25 mg/kg (* *p* < 0.05). No significant differences in IL-1 levels were observed in either group.

#### 2.3.2. Composition of F1

A GC–MS analysis identified five fatty acids and derivatives, shown in Table 5, as the main constituents of the active fraction (F1).

## 3. Discussion

*A. angustifolia* is widely used in Mexico for its nutritional and medicinal properties. Some of the many beneficial effects traditionally attributed to this plant species have been confirmed, especially in conditions with an inflammatory background, such as sprains and psoriasis. An acetonic extract of this plant was reported to show anti-inflammatory activity in a TPA-induced ear edema model [32], with 3-O-((6′-O-palmitoyl)-β-D-glucopyranosyl) sitosterol being identified as the active constituent. This compound induced a dose-dependent effect and was able to modulate local cytokine levels in the ear of mice exposed to the stimulus [33]. Additionally, the oral administration of 200 mg/kg of a methanolic *A. angustifolia* extract inhibited carrageenan-induced plantar edema by 55% in mice [9].

The anti-inflammatory activity of an orally administered acetone extract of *A. angustifolia* (AaAc) and its fractions was herein assessed in a model of ear edema induced by xylene, a volatile and toxic substance that causes skin irritation [34]. Xylene application to mouse ears results in robust edema and long-lasting erythema. Local xylene application has been shown to promote vascular reactions mediated by the activation of the TRPA1 channel [35], which is expressed in mast cells, keratinocytes, melanocytes, and sensory nerves, among other cell types [36,37]. Xylene application causes severe vasodilatation and inflammatory cell infiltration [38], with substance P also reported to mediate its effects [39].

AaAc showed the highest anti-inflammatory activity, followed by F-MeOH and F-Ac, indicating that these products might compensate for some of the mechanisms underlying the irritant activity of xylene. F-Ac resolution yielded four fractions with different chemical compositions. Among them, F1 showed the highest anti-inflammatory effect in the TPA assay; therefore, its capacity was evaluated at different doses in a K/C-induced mono-arthritis model.

GC–MS analysis showed that F1 is composed of a mixture of fatty acids and their esters, including methyl palmitate, ethyl palmitate, ethyl octadecenoate (ethyl stearate), ethyl linoleate, and ethyl oleate. These compounds have shown anti-inflammatory activity in several biological models, including some experimental arthritis models. For instance, methyl palmitate (the methyl ester of palmitic acid) inhibited phagocyte function in a culture of RAW cells when stimulated with LPS; it also decreased TNF-α and increased IL-10 levels, while decreasing the phosphorylation of the inhibitory protein kappa B (IκBα) [27,31]. Both methyl palmitate and ethyl palmitate were found to reduce carrageenan-induced plantar edema in rats, in addition to decreasing PGE2 levels in inflammatory exudates. In another study, methyl palmitate and ethyl palmitate inhibited carrageenan-induced plantar inflammation by decreasing prostaglandin E2 (PGE2) levels in the exudate. Furthermore, in a rat model of endotoxemia, both compounds reduced TNF-α and IL-6 levels by decreasing NF-κB expression in the lung and liver, in addition to counteracting histopathological changes caused by LPS. Both fatty acid esters reduced croton oil-induced ear edema, decreasing neutrophil infiltration by inhibiting myeloperoxidase activity [28]. Moreover, some fractions of a *Jatropha curcas* extract that showed anti-inflammatory activity in RAW264.7 cells contain fatty acids such as octadecanoic acid methyl ester and free octadecanoic acid [40]. Furthermore, when added to the diet of mice with collagen-induced experimental arthritis, linoleic acid was able to offset the damage, reducing joint edema, blood IL-1β levels [41], and hepatic arachidonic acid levels [42].

As mentioned above, the major components of F1 have been proved to modulate the immune response mediated by the proinflammatory cytokines TNF-α, IL-1β, and IL-17, and the regulatory molecule IL-10. On the other hand, it is an established fact that none of these proteins is totally pro- or anti-inflammatory. In this case, IL-10 is a pleiotropic cytokine that exerts regulatory effects through various mechanisms; it is capable, for example, of suppressing the activation and function of immune cells, inhibiting edema and the production of TNF-α, and downregulating the expression of cyclooxygenase-2 (COX-2) and PGE2, among other mediators. IL-10 blocks neutrophil infiltration and synovial tissue activation, prevents macrophage polarization towards the M2 phenotype, and selectively inhibits the expression of pro-inflammatory cytokines, especially IL-1β and TNF-α [24]; both molecules are known to promote the progression of rheumatoid arthritis.

In the carrageenan-induced intraarticular inflammation model studied in this work, animals treated with F1 at all doses consistently showed increased levels of splenic and joint IL-10 with respect to the VEH group. F1, whose main components are fatty acids, could modulate IL-10 release; in turn, this could partially inhibit IL-1β expression. Thus, F1 showed no effect on the levels of that cytokine in the joint, although a slight but significant decrease in IL-1β levels in the spleen was observed at low doses of F1. This modulation could be evidenced by a significant decrease in TNF-α levels in both tissues.

Finally, at doses of 25 and 50 mg/kg, F1 increased the response time in the hot plate assay, suggesting an analgesic action. The imbalance in the central and peripheral mechanisms of pain includes the direct activation or the increment in the sensitivity of the nociceptors to different stimuli, such as joint inflammation, through the elevation of pro-inflammatory cytokines released by immune cells in the synovium, including TNF-α, IL-1β, IL-6, and IL-17, which directly alter the responses of nociceptive neurons [43]. Therefore, the analgesic effect of F1 could be related to a decrease in proinflammatory molecules and, with them, the inflammation and, consequently, the overactivation of nociceptors. In addition, the increase in the concentration of IL-10 in joints, a cytokine that decreases inflammation, reduces fever, acute-phase protein release, and vascular permeability and is an inhibitor of hyperalgesia [44].

F1 is an active mixture of aliphatic-chain fatty acids, capable of reducing local and joint inflammation, which modulates the immune response by blocking the increase in the concentration of pro-inflammatory cytokines and raising IL-10 levels. Therefore, it is necessary to continue with the pharmacological investigation of this mixture of compounds in more complex models of RA, which could lead to the identification of a mode of action for a possible antiarthritic treatment.

## 4. Material and Methods

### 4.1. Plant Material, Extract, and Fraction Preparation

*A. angustifolia* leaves were obtained from a 5-year-old specimen collected in a controlled cultivation field located in Tlalquitenango (Morelos State), at 18°37′ N latitude, 99°10′ W longitude, and an elevation of 911 m above sea level. This culture belonged to the company “Yautli”, which houses the identity register depositaries of the specimens of this species. An herbarium specimen was sent to the herbarium of the Institute of Biology of the National Autonomous University of Mexico and was identified by Dr. Abisaí Mendoza with the number 1,419,710.

Stalks were cut 5 cm from the core (piña). The material was cut into small pieces, frozen, and freeze-dried; the dried material was ground and macerated with acetone for 24 h. The resulting liquid was filtered and concentrated to dryness on a rotary evaporator (R-114, Buchi, Flawil, Switzerland). This procedure was repeated three times to yield the acetone extract (AaAc). The extract was kept at 4 °C until it was used for chemical analysis and in biological tests.

### 4.2. Chemical Analysis

#### 4.2.1. Fractioning the Acetone Extract (AaAc)

Based on previous studies, the acetone extract [33], AaAc (28 g), was absorbed on silica gel in a 2:1 ratio to be placed in a glass column (100 × 400 mm^2^) packed with normal-phase silica gel (200 g, 70–230 mesh, Merck, Kenilworth, NJ, USA). After elution with 1500 mL of hexane as the mobile phase, the solvent was concentrated on a rotary evaporator to yield the hexane fraction (F-Hex). Then, 1500 mL of ethyl acetate was added to the column, which was concentrated to obtain the ethyl acetate fraction (F-AcOEt), and the process was repeated with acetone and methanol as mobile phases to obtain the acetone (F-Ac) and methanolic (F-MeOH) fractions, respectively. These fractions were evaluated in a murine model of xylene-induced local edema [38].

#### 4.2.2. Separation of Acetone Fraction (F-Ac)

F-Ac (1.2 g) was selected for chromatographic separation, given its higher activity. The fraction was absorbed on silica gel (10 g, 70–230 mesh, Merck, Kenilworth, NJ, USA), placed in a glass column (50 × 300 mm^2^) packed with silica gel (80 g), and eluted with a dichloromethane:methanol gradient system as the mobile phase with a 10% increase in polarity, collecting fractional volumes of 100 mL. In total, 30 fractions were obtained, which were then pooled according to similarity via thin-layer chromatography (TLC) into four fractions (F1, F2, F3, and F4).

Due to the low availability of xylene and the yield of fractions, a TPA-induced local edema assay was used to evaluate the anti-inflammatory activity of F1–F4. TPA is a phorbol ester, widely used to study local inflammation by activating PKC with leukocyte infiltration and promoting eicosanoid release [45].

#### 4.2.3. Gas Chromatography–Mass Spectrometry Analysis

The chemical composition of the most active fraction (F1) was determined on a gas chromatograph (GC, Agilent Technology 6890 masses coupled 5973N, Santa Clara, CA, USA) fitted with a quadrupole mass detector in electron impact mode at 70 eV. Volatile compounds were resolved on an HP 5MS capillary column (25 m length, 0.2 mm inner diameter, 0.3 µm film thickness). The oven temperature was set at 40 °C for 2 min, then increased from 40 to 260 °C at a rate of 10 °C/min and maintained at 260 °C for 20 min. The mass detector interface temperature was set to 200 °C and the mass acquisition range was 20–550. Injector and detector temperatures were set at 250 and 280 °C, respectively. Helium was used as the carrier gas, with a flow rate of 1 mL/min. The major compounds in the mixture were identified by comparing their mass spectra with those of the National Institute of Standards and Technology (NIST) Library 1.7 [46].

### 4.3. Biological Tests

#### 4.3.1. Animals

Female BALB/c mice (20–25 g) were acquired from the vivarium of the Centro Médico Nacional Siglo XXI-IMSS. Mice were randomly distributed into groups of 7 animals and were maintained under controlled conditions for at least 3 weeks before the experiments, with a 12 h light/dark cycle at 20 ± 1 °C; food (Labdiet 5008, Brentwood, MO, USA) and water were provided ad libitum. Behavioral tests were performed in an isolated, well-lit room, with a video recording system. Mice were handled in accordance with the Mexican Official Standard for the Care and Handling of Animals (NOM-062-ZOO-1999) and in compliance with all international regulations. The IMSS Research Committee approved the experimental protocol (permit number R-2015-1702-4).

#### 4.3.2. Xylene-Induced Mouse Ear Edema

Each treatment was administered orally (p.o.) to groups of 7 mice 1 h before the application of xylene (irritant) to the pinna. Different groups received AaAc 200 mg/kg, F-Hex, F-AcOEt, F-Ac, or F-MeOH at a dose of 50 mg/kg each; a positive control group received indomethacin (INDO, ≥99% purity by TLC, Sigma Chemical Co., Saint Louis, MO, USA) at a dose of 5.0 mg/kg; and a negative control group was administered with 1% Tween-20 solution (VEH, 100 µL/10 g body weight). The assay was performed as described in [29]. Briefly, 40 µL of xylene was applied to the right pinna, 20 µL to the inner side and 20 µL to the outer side; for reference, 20 µL of 70% ethanol was applied to the inner and outer sides of the left ear. One hour after the irritant stimulus, the animals were sacrificed by cervical dislocation, and 6 mm diameter circular sections were obtained from the right and left ears. Both treated (*t*) and untreated (*nt*) ears were weighed to measure the level of inflammation. The percentage of edema inhibition was calculated as follows:$$\mathrm{Inhibition}\;\left(\%\right)=\left|\frac{\Delta \mathrm{veh}-\Delta_{\mathrm{tr}}}{\Delta \mathrm{veh}}\right|\times 100$$
Δveh = ear weight of VEH groupΔtr = ear weight of treatments

#### 4.3.3. TPA-Induced Mouse Ear Edema

Edema was induced in the ear of mice following a previously described method [46]. A negative control group administered with vehicle (VEH) and a positive group receiving indomethacin (INDO, 1.0 mg/ear dissolved in acetone) as described above were included. Fractions F1–F4 were administered topically at a dose of 1.0 mg/ear dissolved in acetone, 10 µL on the outer and 10 µL on the inner side of the left ear of each mouse, leaving the right ear untreated as a control. Ten minutes later, 2.5 µg/ear of 12-O-tretradecanoylphorbol-13-acetate (TPA, ≥99% purity by HPLC) was administered. Four hours later, the mice were sacrificed by cervical dislocation and 6 mm diameter circular sections were obtained from each ear. Both treated (*t*) and untreated (*nt*) ears were weighed to measure the level of inflammation. The percentage of edema inhibition was calculated as follows:$$\mathrm{Inhibition}\;\left(\%\right)=\left|\frac{\Delta \mathrm{veh}-\Delta_{\mathrm{tr}}}{\Delta \mathrm{veh}}\right|\times 100$$
Δveh = ear weight VEH groupΔtr = ear weight of treatment groups

#### 4.3.4. Kaolin/Carrageenan (K/C)-Induced Mono-Arthritis

Mice were distributed into groups of 8 animals. One arthritis-free group was administered with sterile saline (SS), and the other groups were given an intraarticular kaolin/carrageenan (K/C) challenge on the first day of the experiment, as described below. Based on the previous study [47], F1 was administered p.o. at a dose of 12.5, 25, or 50 mg/kg to the different groups, starting on day 2 and continuing until day 10. A negative control group (VEH) and a positive control group, treated with methotrexate (MTX), were also included.

For the induction of mono-arthritis [48], the base of the left knee joint was measured preoperatively in animals of all groups as a reference, using a digital micrometer (Mod. MDC-1” SB, Mitutoyo Products, Kanagawa, Japan). The animals were then anesthetized with 55 mg/kg sodium pentobarbital intraperitoneally, and 40 µL of 40% kaolin solution was injected into the joint cavity of the left knee; immediately thereafter, a series of flexions and extensions were performed for 15 min. Then, 40 µL of 2% carrageenan solution was injected into the joint cavity, and flexions and extensions were performed again for 5 min. Mice in an untreated group were handled similarly, but SS was injected instead of K/C. Joint edema was measured once daily as described above. Subsequent treatments were administered every 24 h after the initial K/C administration. Mice were sacrificed on day 10; the left knee and spleen were dissected out and stored at −70 °C until use.

#### 4.3.5. Thermal Hyperalgesia Assay

Thermal hyperalgesia was assessed in a hot plate assay. The animals were placed one at a time on a surface at 50 ± 2 °C. The latency time to respond to the stimulus by licking and/or shaking the hind paw or jumping was measured on the last day of the experiment with a hand-held stopwatch, with a maximum time allowed of 20 s [49,50].

#### 4.3.6. Spleen Index

The spleen of each animal was weighed, and the spleen index was calculated as a percentage with respect to the total body weight of the animal [51].

#### 4.3.7. Homogenization of Spleen and Joint Tissues

Joint and spleen samples were triturated with dry ice in a mortar until the dry ice evaporated. Joint and spleen tissues were placed in a vial with 2 mL of PBS, pH 7.4, and 0.01% PMFS in isopropyl alcohol. The tissues were homogenized (Mod. D-500 Pack 1, Dragon Lab, Meriden, CT, USA) for 10–15 s and centrifuged at 12,000 RPM for 5 min. Five 300 µL aliquots of supernatant were stored at −70 °C until cytokine quantification.

#### 4.3.8. Quantification of Anti-Inflammatory and Pro-Inflammatory Cytokines

Cytokines were quantified by ELISA using a commercial kit (OptEIATM, BD Biosciences, Franklin Lakes, NJ, USA) following the manufacturer’s instructions. Briefly, 100 µL/well of the capture antibody were placed in 96-well plates and incubated for 12 h at 4 °C. Then, the plates were washed three times with 300 µL/well of 0.05% Tween-20 in PBS. The plates were supplemented with 100 µL of PBS containing 10% fetal bovine serum (FBS), pH 7.0, and left to stand for 1 h at room temperature. The contents were discarded, and the plate was washed three times with 300 µL/well of 0.05% Tween-20 in PBS. Then, 100 µL of standard, target (PBS with FBS), and sample was added. The plates were incubated for 2 h at room temperature. The contents were discarded, and the plates were washed five times with 300 µL/well of 0.05% Tween-20 in PBS. For TNF-α, IL-6, IL-4, and IL-10 quantification, 100 µL/well of streptavidin-HRP-conjugated detection antibody was added. The plates were incubated for 1 h and washed seven times with 300 µL/well of 0.05% Tween-20 in PBS.

For IL-1β quantification, 100 µL/well of antibody detection was added, incubated for 1 h at room temperature and washed five times with 300 µL/well of 0.05% Tween-20 in PBS, followed by 100 µL/well of streptavidin-HRP. The plates were incubated at room temperature for 1 h and washed seven times with 300 µL/well of 0.05% Tween-20 in PBS.

Then, 100 µL/well of o-phenylenediamine substrate was added, and the plates were incubated for 30 min at room temperature in the dark. The reaction was stopped with 2N H_2_SO_4_, and the plates were read in a Stat Fax 2100 spectrophotometer (Awareness Technologies, Bellport, NY, USA) at 450 nm and 37 °C.

For IL-17 quantification, 50 µL/well of RD1–38 or a standard was added to the plates. The plates were gently shaken for 1 min and incubated at room temperature for 2 h. The contents were discarded, and the plates were washed five times with PBS. Then, 100 µL of HBR-conjugated antibody was added, incubated for 2 h at room temperature, and washed five times. Subsequently, 100 µL of substrate solution was added, and the plates were incubated for 30 min in the dark. The reaction was stopped with 2N H_2_SO_4_, and the plates were read at 450 nm and 37 °C.

### 4.4. Statistical Analysis

Data were analyzed with SPSS v.22.0 software, using one-way ANOVA followed by a Bonferroni test. Differences with respect to the negative control were regarded as significant for *p* ≤ 0.05. Data are reported as mean ± SD.

## 5. Conclusions

*Agave angustifolia* is an edible plant to which various medicinal properties have been attributed. Its value as an anti-inflammatory and immunomodulatory remedy could be due to its content of free fatty acids and their esters. The extract herein evaluated showed a clear anti-inflammatory effect in TPA- and carrageenan-induced arthritis models. Interestingly, the extract also showed an analgesic effect, which should be further studied in other models. It is necessary to continue with the pharmacological study of F1 and the fatty acids individually, through histology and in vitro studies (culture cells). This information will allow future researchers to describe the mode of action of the antiarthritic effect (anti-inflammatory and immunomodulatory) and propose this plant as a possible antiarthritic drug.

## Figures and Tables

**Figure 1 molecules-27-07204-f001:**
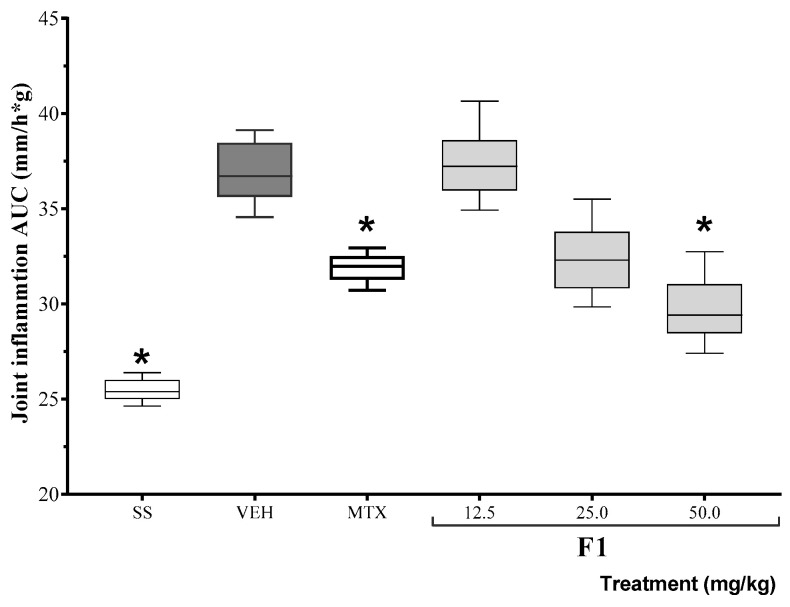
Effect of the F1 fraction of *A. angustifolia* extract at different doses on the AUC of the time course plot of joint inflammation in a murine model of K/C-induced mono-arthritis. The plots show the mean and SD for each group (*n* = 8). Asterisks indicate significant differences with respect to the vehicle group, administered with 1% Tween-20 (VEH), * *p* ≤ 0.05, by ANOVA with post hoc Bonferroni test. SS: saline solution; MTX: methotrexate.

**Figure 2 molecules-27-07204-f002:**
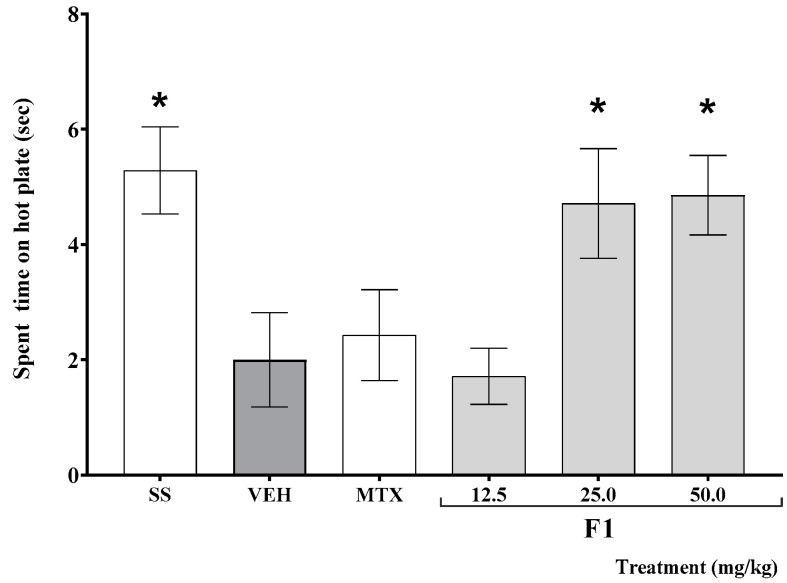
Effect of the F1 fraction of *A. angustifolia* extract at different doses on thermal hyperalgesia in inflamed joint in a murine K/C-induced mono-arthritis model. Data are reported as the mean ± SD for each group (*n* = 12) and were analyzed by ANOVA with post hoc Bonferroni test. * *p* ≤ 0.05 with respect to the vehicle-treated group (Tween-20, VEH). SS: saline solution; MTX: methotrexate.

**Table 1 molecules-27-07204-t001:** Effect of the extract and fractions from *A. angustifolia* on xylene-induced ear edema in mice.

Treatment (mg/kg)	Edema (mg)	Inflammation Inhibition (%)
VEH	8.87 ± 2.8	-
IND (5.0)	2.40 ± 1.8 *	72.9
AaAc (200.0)	1.52 ± 0.8 *	82.9
**Fractions**		
F-Hex (50.0)	5.36 ± 0.6 *	39.6
F-AcOEt (50.0)	5.10 ± 2.2 *	42.6
F-Ac (50.0)	4.75 ± 1.6 *	46.4
F-MeOH (50.0)	4.50 ± 2.1 *	49.3

Data are reported as mean ± standard deviation (SD) and were analyzed by analysis of variance (ANOVA) with post hoc Bonferroni test (*n* = 7). * *p* < 0.05 with respect to the negative control group. IND = indomethacin.

**Table 2 molecules-27-07204-t002:** Effect of the extract and fractions from *A. angustifolia* on TPA-induced ear edema.

Treatment (mg/ear)	Edema (mg)	Inflammation Inhibition (%)
VEH	10.24 ± 2.10	-
IND (1.0)	1.65 ± 0.21 *	83.8
F1 (1.0)	3.06 ± 2.00 *	70.1
F2 (1.0)	7.87 ± 0.23 *	23.1
F3 (1.0)	6.58 ± 0.51 *	35.7
F4 (1.0)	7.08 ± 1.20 *	30.8

Data are reported as mean ± SD and were analyzed by ANOVA with post hoc Bonferroni test (*n* = 7). * *p* < 0.05 with respect to the negative control group. IND = indomethacin.

**Table 3 molecules-27-07204-t003:** Effect of the F1 fraction of *A. angustifolia* extract on spleen index (percentage of total body weight).

Treatment (mg/kg)	Relative Weight of Spleen (%)
SS	0.552 ± 0.045 *
VEH	0.638 ± 0.035
MTX (1.0)	0.582 ± 0.050 *
F1 (12.5)	0.573 ± 0.043 *
F1 (25.0)	0.632 ± 0.030
F1 (50.0)	0.587 ± 0.044 *

Data are reported as mean ± SD and were analyzed by ANOVA with post hoc Bonferroni test (*n* = 7). * *p* < 0.05 with respect to the negative control group. SS: saline solution; VEH: Tween-20; MTX: methotrexate.

**Table 4 molecules-27-07204-t004:** Effect of the extract and fractions from *A. angustifolia* on cytokine levels in spleen and left knee joint in a K/C-induced mono-arthritis model.

Treatment (mg/kg)	Cytokine Levels (pg/mg Protein)			
	IL-1β	TNF-α	IL-17	IL-10
Spleen				
SS	15,589.6 ± 3110.3 *	10,339.97 ± 2379.4 *	4588.2 ± 253.6 *	17,474.8 ± 1768.0 *
VEH	20,661.5 ± 1302.9	23,015.1 ± 4134.2	5917.7 ± 997.5	7742.0 ± 3688.6
MTX (1.0)	18,360.4 ± 1396.1	11,751.53 ± 3765.9 *	5090.7 ± 767.7	19,709.9 ± 3429.0 *
F1 (12.5)	18,070.5 ± 596.1 *	11,547.29 ± 1792.1 *	5576.8 ± 916.1	17,271.0 ± 625.0 *
F1 (25.0)	17,639.1 ± 409.7 *	12,296.05 ± 1541.5 *	6291.1 ± 669.0	16,670.3 ± 370.5 *
F1 (50.0)	18,481.0 ± 1381.8	11,844.34 ± 1669.6 *	5212.9 ± 738.2	17,142.5 ± 1728.8 *
Left knee joint				
SS	13,883.5 ± 2957.9	6808.08 ± 2761.4 *	1258.7 ± 261.5 *	15,938.4 ± 1855.6 *
VEH	18,382.1 ± 3616.2	10,762.80 ± 564.4	2188.5 ± 249.1	9832.6 ± 1514.5
MTX (1.0)	12,242.5 ± 2605.4	8080.48 ± 1703.3 *	1561.2 ± 324.1 *	13,501.1 ± 1382.2 *
F1 (12.5)	18,807.9 ± 1401.3	9569.87 ± 1038.7	2337.4 ± 168.6	15,803.1 ± 1362.1 *
F1 (25.0)	17,465.7 ± 1398.0	9145.84 ± 588.6 *	1556.8 ± 196.8 *	15,506.8 ± 1075.4 *
F1 (50.0)	15,112.1 ± 780.9	9064.67 ± 313.6 *	1713.1 ± 232.5	14,851.6 ± 989.2 *

Data are reported as mean ± SD and were analyzed by ANOVA with post hoc Bonferroni test (*n* = 7). * *p* < 0.05 with respect to the negative control group. SS: saline solution; VEH: Tween-20; MTX: methotrexate.

**Table 5 molecules-27-07204-t005:** Chemical composition of F1, according to GC–MS analysis.

RT (min)	Fatty Acid Name	Composition (%)	General Data
18.71	Palmitic acid, methyl ester (methyl palmitate)	20	A C16 saturated fatty acid, a natural product with a vasodilator effect on rat/rabbit thoracic aorta [26]; anti-inflammatory activity in different assays [27,28]
19.32	Palmitic acid, ethyl ester (ethyl palmitate)	10	Ethyl hexadecanoate is a long-chain fatty acid ethyl ester resulting from the formal condensation of the carboxy group of palmitic acid with the hydroxy group of ethanol; it has a role as a plant metabolite; anti-inflammatory activity in different assays [27,28,29]
20.43	Octadecenoic acid, methyl ester (methyl octadecenoate or ethyl stearate)	35	This aliphatic-chain fatty acid had a neuroprotective effect on dopaminergic cells in a model of Parkinson’s disease [30]
20.93	Linoleic acid, ethyl ester (ethyl linoleate)	13	An essential fatty acid, which among other things reduces the levels of nitric oxide and PGE2 by downregulating the enzymes involved in their production (iNOS and COX-2) [31]
20.98	Oleic acid, ethyl ester (ethyl oleate)	8	

## Data Availability

Not applicable.

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
