# Peer review of "Fatty-Acid-Rich Agave angustifolia Fraction Shows Antiarthritic and Immunomodulatory Effect"

_molecules, 2022, doi:10.3390/molecules27217204_

Round 1

Reviewer 1 Report

In this manuscript the authors describes a promising   antiarthritic and immunomodulatory fatty acid rich  fraction of Agave anguistifolia . The active fraction designed as F1 is a sub-fraction of chromate graphic acetone fraction of crude acetone extract of leaves. This is quiet interesting and authors could document the fatty acid components of this active fraction. The manuscript has been written well well with adequate details of methodology. Discussion part need to have concluding paragraph at the end. 

·         A major concern regarding the manuscript is the logic of the study which has not been stated clearly in the Introduction part.

·         Novelty is also lacking as the same author has published a manuscript (Hernández-Valle et al (2014)Anti-Inflammatory Effect of 3-O-[(6'-O-Palmitoyl)-β-d-glucopyranosyl Sitosterol] from Agave angustifolia on Ear Edema in Mice,. Molecule, 19(10): 15624–15637) in which anti inflammatory molecules has been isolated from the stem of the same plant. The active fraction isolated was Again acetone fraction and the active molecules documented were sitosterols. The question is why the authors  give emphasise to leaves fatty acid  derivatives  in studying anti arthritic potential?

·         The title of manuscript contain the term immune modulatory effect. What definite experiment has been carried out to show its immunomodulatory efficacy?

The manuscript need revision

Reviewer 2 Report

The manuscript reported the biologic activities of extracts of Agave angustifolia. The results showed  spleen and joint cytokine levels could be modulated and thus decreased pain. However, how that correlated to the consequence of pain relieving should be clearly elaborated. The statistical analysis of data needs to be refined to show the significant difference between the control and the tested subjects. According to an GC-MS analysis, the main components of F1 are fatty acid derivatives, which need to be further tested to show the anti-inflammatory activities. The GC-MS analysis can be improved to unravel the composition of active ingredients of the fractions. 

Reviewer 3 Report

Please find my comments on the article Fatty acid-rich Agave angustifolia fraction shows antiarthritic and immunomodulatory effect for revision. Authors needs to make all the suggested changes and requested to highlight them in the manuscript.

1. The introduction should be improved by incorporating a more detailed outline on the biological and pharmacological properties of the plant (line 58 -65)

2. Likewise, the description on RA should also be enriched with recent information on the disease.

3. Also the citations in introduction needs to be increased (several lines require citations)

4. In Figure 2, the margin of bar representing MTX seems to be thicker than others. Make necessary correction

5. The reason for choosing the cytokines (IL-1β, TNF-α, IL-17, IL-10) must mentioned in introduction (emphasizing their role in RA)

6. The chemical structure of individual compounds are not necessary; instead if the authors can provide the % composition (if available)

7. The authentication details of the plant must be provided in the methodology

8. What was the reason for choosing acetone extract? Support of previous literature may be given, if avaialble

9. What was the reason for choosing the strain and sex of mice?

10. The entry of formula may be done with the help of "Equation" option under "Insert" in MS Word

11. Conclusion must contain the future aspects of the study.

12. How the doses were determined? Any information on the toxicity study?

Round 2

Reviewer 1 Report

The authors have satisfactorly addressed to my comments in the revised version. I do not any further suggestion

Reviewer 3 Report

The manuscript has been revised according to the comments. No more comments to the authors